# Nd^3+^, Yb^3+^:YF_3_ Optical Temperature Nanosensors Operating in the Biological Windows

**DOI:** 10.3390/ma16010039

**Published:** 2022-12-21

**Authors:** Maksim Pudovkin, Ekaterina Oleynikova, Airat Kiiamov, Mikhail Cherosov, Marat Gafurov

**Affiliations:** Institute of Physics, Kazan Federal University, 18th Kremlyovskaya Street, Kazan 420008, Russia

**Keywords:** lifetime thermometry, Nd^3+^, Yb^3+^:YF_3_, optical temperature sensors

## Abstract

This work is devoted to the study of thermometric performances of Nd^3+^ (0.1 or 0.5 mol.%), Yb^3+^ (X%):YF_3_ nanoparticles. Temperature sensitivity of spectral shape is related to the phonon-assisted nature of energy transfer (PAET) between Nd^3+^ and Yb^3+^). However, in the case of single-doped Nd^3+^ (0.1 or 0.5 mol.%):YF_3_ nanoparticles, luminescence decay time (LDT) of ^4^F_3/2_ level of Nd^3+^ in Nd^3+^ (0.5 mol.%):YF_3_ decreases with the temperature decrease. In turn, luminescence decay time in Nd^3+^ (0.1 mol.%):YF_3_ sample remains constant. It was proposed, that at 0.5 mol.% the cross-relaxation (CR) between Nd^3+^ ions takes place in contradistinction from 0.1 mol.% Nd^3+^ concentration. The decrease of LDT with temperature is explained by the decrease of distances between Nd^3+^ with temperature that leads to the increase of cross-relaxation efficiency. It was suggested, that the presence of both CR and PAET processes in the studied system (Nd^3+^ (0.5 mol.%), Yb^3+^ (X%):YF_3_) nanoparticles provides higher temperature sensitivity compared to the systems having one process (Nd^3+^ (0.1 mol.%), Yb^3+^ (X%):YF_3_). The experimental results confirmed this suggestion. The maximum relative temperature sensitivity was 0.9%·K^−1^ at 80 K.

## 1. Introduction

In our time, technological needs in various fields have reached such a development that conventional contact temperature sensors can no longer perform accurate measurements with submicrometer spatial resolution [1,2,3]. Traditional methods of temperature measuring are thermocouples, thermistors, and infrared cameras. They are not able to provide high spatial resolutions (rough estimation ~λ/2) and/or the required contactlessness. So, the development of new non-contact temperature sensors is mandatory in modern science and industry. In this case, the luminescence temperature sensing and/or mapping satisfies the above-mentioned requirements. In this method, the temperature determination can be performed by analyzing a temperature-dependent luminescence signal (in the majority of cases, these parameters are luminescence intensity, lifetime, and band shape) of the nano- or micro-sized phosphors which are in contact with the studied object. In its turn, working in the UV, visible and/or NIR spectral ranges allows obtaining submicrometer spatial resolution. Luminescent thermometry is highly required in medicine and biology [4,5] for thermography, for the clinical diagnosis of cancerous tumors [6], as well as for measuring the temperature of integrated circuits and micro devices in order to check their stability and proper functioning [1,2]. In this work, phosphors based on fluoride host doped with rare-earth ions are used. This class of materials has a high chemical stability, mechanical strength, sufficiently high melting point, relatively low probability of non-radiative processes [7], high quantum yield of luminescence [8], and low toxicity [9]. It should also be noted, that the modern methods of synthesis allows obtaining fluoride phosphors with desirable morphology and structure [10]. YF_3_ host was chosen because of the low energy of phonons (500 cm^−1^), and, consequently, the low probability of multi-phonon non-radiative relaxation. Nd^3+^ and Yb^3+^ doping ions were chosen because of the possibility of complex temperature-dependent energy exchange probabilities between the ions, which can provide high temperature sensitivity. Also, the excitation and emission wavelengths are situated in the near IR range (tissue transparency window), which is very important for biomedical applications [11]. Down-conversion optical temperature sensors based on Nd^3+^/Yb^3+^ ion pair were recently studied in works [12,13,14]. In these phosphors, the emission of Yb^3+^ is observed under the Nd^3+^ excitation revealing the energy transfer between Nd^3+^ and Yb^3+^. The luminescence intensity ratio (*LIR*) between Nd^3+^ emission (^4^F_3/2_–^4^I_9/2_ transition at ~866 nm) and Yb^3^+ emission (^2^F_5/2_–^2^F_7/2_ transition at ~980 nm) can be taken as a temperature-dependent parameter. The main mechanism of temperature sensitivity is related to the phonon-assisted nature of energy transfer between Nd^3+^ and Yb^3+^. In the literature, there are three main energy transfer processes between the above-mentioned ions that can be characterized by their probabilities: W_ET_—probability of energy transfer from Nd^3+^ to Yb^3+^, W_BET_—probability of back energy transfer from Yb^3+^ to Nd^3+^, W_DIFF_—probability of energy diffusion between Yb^3+^ ions [15]. In the case of a low concentration of Nd^3+^ energy transfer from Nd^3+^ to Nd^3+^ is considered to be negligible. The probabilities of above-mentioned energy transfer processes are competitive and they are dependent on doping ion concentrations. In particular, W_BET_ decreases with the increase of Yb^3+^ concentration. This phenomenon paves the way toward the manipulation of temperature sensitivity via Yb^3+^ concentration. Indeed, it was experimentally demonstrated in the works [12,14,16]. However, it seems, that the Nd^3+^/Yb^3+^ doped phosphors are studied for relatively high Yb^3+^ concentrations (>1 mol.%). Specifically, in our previous work it was shown, that the emission of the sample Nd^3+^ (0.5%), Yb^3+^ (0.5%):YF_3_ is negligible under Nd^3+^ excitation (355 nm, ^4^I_9/2_–^4^D_3/2_ absorption band of Nd^3+^) and this sample was not studied. On the other hand, it can be suggested, that the excitation conditions (mostly excitation wavelength) could affect the relative emission intensities of both Nd^3+^ and Yb^3+^ and the intense Yb^3+^ emission can be observed for the samples, containing 1.0 mol.% of Yb^3+^ and even less. Indeed, the Nd^3+^/Yb^3+^-based optical sensor with lower Yb^3+^ concentrations (<1.0 mol.%) can demonstrate higher *S_r_* and S_a_. One of the highest *S_r_* values obtained for Nd^3+^ (0.5%), Yb^3+^ (8.0%):YF_3_ (~0.6%/K at 144 K) [12], Nd^3+^ (0.5%), Yb^3+^ (5.0%):LiLaP_4_O_12_ (~0.3%/K at 300 K) [15]. The double-doped inorganic nano- or microparticles are capable of demonstrating higher performances compared to their single-doped counterparts in the relatively broad temperature range of 10–400 K. Specifically, Pr^3+^, Yb^3+^:LaF_3_ down-conversion nanoparticles demonstrate one of the highest sensitivities (*S_r_* ~ 6.0%·K^−1^ at 10 K) in the cryogenic temperature range [17]. The high temperature sensitivity is attributed to the convenient phonon-assisted energy transfer as well as quantum cutting phenomenon. In its turn, Nd^3+^, Yb^3+^:YPO_3_ and Nd^3+^, Yb^3+^:LaPO_3_ phosphors are capable of reaching 1.2 and 1.0%·K^−1^, respectively at 300 K [18]. Here, the high temperature sensitivity is explained by the fact, that the efficiency of Yb^3+^ back energy transfer to Nd^3+^ increases with the increase of temperature that leads to the faster change of spectral-kinetic characteristics with temperature. In the case of Tm^3+^, Yb^3+^:LiYF_4_ down-conversion phosphors, they demonstrate maximum temperature sensitivities around 1.2%·K^−1^ at 300 K. The high performance is explained by phonon-assisted energy transfer from ^3^H_4_ (Tm^3+^) to ^2^F_5/2_ (Yb^3+^). Note, that Tm^3+^, Yb^3+^ system has one of the biggest energy gap between interacting ^3^H_4_ (Tm^3+^) and ^2^F_5/2_ (Yb^3+^) levels around 2000 cm^−1^.

The objective of this work was to study physical background of functioning of the Nd^3+^, Yb^3+^:YF_3_ luminescent temperature sensors having different concentrations of doping ions. The tasks of the work were to investigate of the spectral-kinetic characteristics of Nd^3+^ and to calculate of the main characteristics of temperature sensors. In particular, the temperature evolution of luminescence spectra will be studied. The luminescence intensity ratio (*LIR*) of Nd^3+^ and Yb^3+^ emissions at different temperatures will allow concluding about energy exchange processes between Nd^3+^ and Yb^3+^ ions. The temperature evolution of luminescence decay curves of single-doped Nd^3+^:YF_3_ and double-doped Nd^3+^, Yb^3+^:YF_3_ will also allow concluding about energy exchange processes between the ions as well as concluding about the contribution of other temperature-dependent processes related, for example, to the thermal expansion of crystal lattice.

We guess that the main novelty of the work is that the work deals with physical background of temperature sensitivity of spectral-kinetic characteristics of the studied Nd^3+^, Yb^3+^:YF_3_ phosphors. However, most recent articles do not take into consideration the thermal expansion phenomenon. In the present paper, we make a hypothesis of temperature sensitivity based on conventional knowledge about phonon-assisted nature of energy transfer between Nd^3+^ and Yb^3+^ as well as on thermal expansion by demonstrating the lattice parameters change with the temperature.

## 2. Materials and Methods

Nd^3+^ (0.1 or 0.5 mol.%), Yb^3+^ (X%):YF_3_ (X = 0.5 and 1.0, 2.0, and 3.0 mol.%) nanoparticles were synthesized via a co-precipitation method with subsequent hydrothermal treatment (180 °C for 30 h) and annealing in vacuum at 500 °C for 5 h. The hydrothermal treatment is used to exclude different ammonium salts and form a pure single-phase YF_3_ doped sample. The annealing procedure improves crystallinity. For Nd^3+^ (0.5%), Yb^3+^ (8.0%):YF_3_ 3.5045 g of Y(NO_3_)_3_·6H_2_O, 0.0219 g of Nd(NO_3_)_3_·6H_2_O, 0.3737 g of Yb(NO_3_)_3_·6H_2_O, were dissolved in 80 mL of distilled water. Then the solution pH was adjusted to 2 with nitric acid. Thereafter, a water solution of NH_4_F (1.4815 g of NH_4_F was dissolved in 10 mL of distilled water) was added dropwise to the mixture while stirring on a magnetic stirrer (400 rpm). In the next step, the solution was treated by hydrothermal synthesis at 180 °C for 30 h. The precipitate was purified with distillated water by centrifugation. The resulting nanoparticles were dried in air at room temperature in a dustproof box. Then the phosphors were annealed in a vacuum at 500 °C for 5 h. The doping ion concentrations are represented in molar percentage (mol.%).

Morphology and size of the samples were studied via transmission electron microscope Hitachi HT7700 Exalens. Sample preparation: 10 microliters of the suspension were placed on a formvar/carbon lacey 3 mm copper grid; drying was performed at room temperature. After drying, the grid was placed on a transmission electron microscope using a special holder for microanalysis. The analysis was held at an accelerating voltage of 100 kV in TEM mode. We built a particle size distribution histogram via commonly used “Image J” software. Since the shape of the particles is not perfectly regular, we calculated the area (squire nanometers) of each particle via 2D TEM image taking into consideration the scale bar. Then we equaled the value of the area to π·D^2^/4 (the area of a circle) and extracted the D values. The statistics are based on the analysis of 100 particles. This method is useful in order to estimate the average size of non-spherical particles. The size distribution histogram was plotted in OriginPro 9.0 software. The size distribution histogram was fitted by LogNormal peak function from OriginPro 9.0 database that is commonly used for particle size analysis.

The phase composition of the particles was studied by means of X-ray diffraction method (XRD) using Bruker D8 ADVANCE X-ray diffractometer (Cu K_α_ radiation, λ = 0.154 nm) having Anton-Paar TTK 450 cooling chamber. We used liquid nitrogen as a cooling agent. The lattice parameter values were calculated in MAUD software (Material Analysis Using Diffraction). The luminescence spectra were recorded via a CCD spectrometer (StellarNet) (0.5 nm spectral resolution). The optical excitation of was performed via IR LD (λ_ex_ = 790 nm, to ^4^I_9/2_–^4^F_5/2_, Nd^3+^ absorption band). The radiation was modulated by rectangular pulses with a period T = 30 ms and pulse duration τ = 5 ms. The experiments were performed in the 80–320 K temperature range via so-called “cold finger” method. The temperature control was carried out via thermostatic cooler “CRYO industries” having LakeShore Model 325 (Westerville, OH, USA) temperature controller. The luminescence decay time curves were recorded via BORDO 211A digital oscillograph (10 bit and 200 MHz bandwidth), MDR-3 monochromator, and photomultiplier PEM-62 (working spectral range ~600–1200 nm). The power density of excitation irradiation was measured with PULSAR-2 powermeter using StarLab software. All the calculations were carried out via Origin.Pro.9.0 software.

In the optical experiments, it is important to avoid the heating of the sample. For this aims, the optimized power density of the excitation irradiation should be chosen. Here, this choice was based on the fact that the shape of the Nd^3+^ peak (850–910 nm, ^4^F_3/2_–^4^I_9/2_ transition) in fluoride hosts (YF_3_ [13], LaF_3_ [19], NaYF_4_ [20]) is temperature-dependent. Hence, before each experiment, we recorded the Nd^3+^ peak (~845–925 nm) for several values of the laser irradiation power density. The luminescence spectra recorded at different values of power density are represented in (Supplementary Information Appendix A). The chosen power density was ~1300 W/m^2^. It should be noted, that due to the use of pulse laser irradiation, the values of the power density are averaged.

## 3. Results and Discussion

### 3.1. Characterization of Nd^3+^, Yb^3+^:YF_3_ Phosphors

A transmission electron microscopy (TEM) image of the Nd^3+^ (0.1%):YF_3_ particles and a size distribution histogram are represented in Figure 1a,b, respectively.

According to the TEM image, the particles have relatively irregular shape reminding of the rhombus. The size distribution histogram is not perfectly fitted by any peak function, probably, due to the non-spherical shape of the particles. The LogNormal fitting determined 231 ± 8 nm average diameter. The width of the size distribution histogram is around 130 nm. We also calculated the common average diameter of the particles (sum of sizes divided by the number of particles) that was equal to 201 nm. Anyway, the size of the particle is larger than 30 nm, hence, the influence of surface can be neglected [21]. Indeed, according to this work, the main unique difference between nanosized crystals and bulk ones is than the number of ions (here Re^3+^ and F^−^) located on the surface of the nanoparticles and the number of ions located in the nanoparticle volume are comparable. The rare-earth ions located on the nanoparticle’s surface have different ligand surrounding compared to rare-earth ion inside the volume. The different surrounding leads to different spectral-kinetic properties. However, according to this work, for rare-earth trifluorides, for nanoparticles larger than 15 nm, the surface ions do not make a serious contribution in the spectral-kinetic properties in opposite to volume ions and nanoparticles are more similar to bulk crystals in term of spectral-kinetic properties. The phase composition of the YF_3_ doped particles was confirmed via XRD. In particular, XRD pattern of Nd^3+^ (0.1%):YF_3_ sample detected for 100, 200, and 300 K and YF_3_ simulation are represented in Figure 2a,b, respectively. The XRD patterns agree with both the simulation and the reference pattern from the Inorganic Crystal Diffractions Database of orthorhombic YF_3_ (P_nma_ space group (no. 074–0911)).

It can also be seen, that some XRD peaks shift toward higher angels with the temperature decrease that can be related to the lattice parameters decrease. Indeed, the calculated *a* constant values were 6.1300(2), 6.2109(3), and 6.3412(1) for 100, 200, and 300 K, respectively. The common shift of the nanoparticle XRD and the simulation of YF_3_ XRD can be related to the presence of doping ions and captured water during the water-based synthesis procedure. It can be seen, that the lattice parameter change is in 0.1 nm/100K range. It can be suggested, that the distance between doping ions changes in the same range. Since, the nature of interaction between the doping ions is dipole-dipole, its efficiency is inversely proportional to r^6^, where r is the distance between interacting ions. It can be suggested, that such relatively small change in distances between the ions can affect the efficiency of interaction between them. For further development of this hypothesis we carried out spectral-kinetic characterization of both single-doped and double-doped samples.

An energy level diagram of the Nd^3+^/Yb^3+^ ion pair is represented in Figure 3**.** The excitation wavelength λ_ex_ = 790 nm corresponds to ^4^I_9/2_–^4^F_5/2_ absorption band of Nd^3+^.

As it was mentioned above, there are three main energy transfer processes between the doping ions that can be characterized by their probabilities: W_ET_—probability of energy transfer from Nd^3+^ to Yb^3+^, W_BET_—probability of back energy transfer from Yb^3+^ to Nd^3+^, W_DIFF_—probability of energy diffusion between Yb^3+^ ions [15,18]. The energy transfer processes for Nd^3+^→Yb^3^+ and for Yb^3+^→Nd^3+^ are accompanied by emission or absorption of phonons, respectively. It provides the temperature sensitivity of the spectral-kinetic properties of the studied samples. In turn, the Nd^3+^ ions interact between each other via cross-relaxation mechanism (^4^F_3/2_–^4^I_15/2_ and ^4^I_9/2_–^4^I_15/2_) that can also affect the temperature sensitivity.

### 3.2. Temperature Dependent Spectral-Kinetic Characterization of Single-Doped Nd^3+^:YF_3_ Nanoparticles and Microparticles

As we mentioned above, the cross-relaxation can also affect the temperature sensitivity. In order to exclude the cross-relaxation process, we synthesized a series of single-doped Nd^3+^:YF_3_ samples having 0.1, 0.5, and 1.0 mol.% concentrations. As we mentioned above, the interplanar distances decrease with the decrease of temperature according to the XRD. Since, the cross-relaxation process is not phonon-assisted, it can be suggested, that the efficiency of cross-relaxation increases with the temperature decrease due to the fact that the distance between Nd^3+^ ions also decreases. The luminescence decay time curves detected at 100, 200, and 300 K for single-doped Nd^3+^ (0.1 (a), 0.5 (b), and 1.0 (c) mol.%):YF_3_ nanoparticles are represented in Figure 4.

In can be seen, that for both Nd^3+^ (0.5 and 1.0 mol.%):YF_3_ samples, decay time decreases with the temperature decrease. The curves can be well described by a single-exponential function. In particular, for Nd^3+^ (1.0 mol.%):YF_3_ sample the decay times were 277, 328, and 358 μs at 100, 200, and 300 K, respectively. In its turn, for Nd^3+^ (0.5 mol.%):YF_3_ sample, the decay times were 370, 389, and 412 μs at 100, 200, and 300 K, respectively. The Nd^3+^ (0.1 mol.%):YF_3_ sample demonstrates ~472 μs decay time. The decrease of decay time with the increase of Nd^3+^ concentration can be explained by concentration quenching phenomenon.

It can be proposed, that at 0.5 mol.% the cross-relaxation between Nd^3+^ ions takes place in contradistinction from 0.1 mol.% Nd^3+^ concentration. The decrease of decay times with temperature can be explained by the decrease of distances between Nd^3+^ with temperature (thermal expansion phenomenon) that leads to the increase of cross-relaxation efficiency. It can be concluded, that for Nd^3+^ (0.1%):YF_3_ nanoparticles the excitation energy does not scatter between Nd^3+^ ions. Since, the (1.0 mol.%):YF_3_ showed the lowest signal-to-noise ratio, we chose 0.1 and 0.5% Nd^3+^ concentration in order to synthesize double doped Nd^3+^, Yb^3+^:YF_3_ and compare their performances.

### 3.3. Temperature Dependent Spectral-Kinetic Characterization of Double-Doped Nd^3+^, Yb^3+^:YF_3_ Nanoparticles

To obtain high temperature sensor performances including relative (*S_r_*) temperature sensitivity, we synthesized a series of Nd^3+^ (0.1 mol.%), Yb^3+^ (0.5, 1.0, 2.0, and 3.0 mol.%):YF_3_ samples. The choice of Nd^3+^ concentration is based on above-mentioned conclusions, that the excitation energy is not scattered between Nd^3+^ ions at 0.1 mol.% concentration. In turn, in order to provide comparable intensities of both Nd^3+^ and Yb^3+^ emissions the Yb^3+^ concentration was varied in the 0.5–3.0 mol.% range. At higher Yb^3+^ concentrations, its emission intensity was much higher than Nd^3+^ emission that led to deterioration of the performances. Indeed, Nd^3+^ (0.1 mol.%), Yb^3+^ (4.0 mol.%):YF_3_ sample demonstrates almost negligible Nd^3+^ emission compared to Yb^3+^ one (Appendix A of the Supplementary File). Normalized at 903 nm (Nd^3+^ emission peak) spectra of Nd^3+^ (0.1 mol.%), Yb^3+^ (1.0 mol.%):YF_3_ sample recorded in the 80–320 K temperature range are represented in Figure 5.

It can be seen, that the spectral shape is notably dependent on temperature. In particular, the Yb^3+^ intensity increases with temperature increase compared to Nd^3+^ emission. It can be explained by the fact that the efficiency of phonon-assisted energy transfer from Nd^3+^ (^4^F_3/2_) to Yb^3+^ (^2^F_5/2_) increases with the temperature increase. Luminescence intensity ratio (*LIR*) between ^4^F_3/2_–^4^I_9/2_ (Nd^3+^) and ^2^F_5/2_–^2^F_7/2_ (Yb^3+^) was taken as a temperature-dependent parameter. The *LIR* curves as functions of temperature are represented in Figure 6.

It can be seen, that all the luminescence intensity ratio (*LIR*) curves demonstrate decreasing behavior. It means, that Yb^3+^ intensity increases faster than Nd^3+^ one with the temperature increase. Such behavior of both intensities reflects the phonon-assisted nature of the energy transfer. Indeed, the efficiency of population of ^2^F_5/2_ (Yb^3+^) increases with the temperature increase as well as depopulation of ^4^F_3/2_ (Nd^3+^). The slight difference in *LIR* functions requires addition study, however, it can be suggested, that this difference related to back energy transfer from Yb^3+^ to Nd^3+^ that is different for different Yb^3+^ concentrations. Indeed, as we mentioned above, under Nd^3+^ excitation, the Yb^3+^ ions can obtain excitation energy. Further, there are at least three processes: Yb^3+^ emits the energy, it can transmit it back to Nd^3+^, and Yb^3+^ can transmit it to Yb^3+^ (energy diffusion). These processes are competitive. In particular, the energy diffusion probability between Yb^3+^ ions increases with the increase of Yb^3+^ concentration. It also decreases the probability of back energy transfer to Nd^3+^. For samples, having different concentrations of doping ions, the ratio between these probabilities is different that leads to difference in the shape of *LIR* curves. The estimation of contribution of the above-mentioned processes is one of the next steps of the present work. In order to calculate *S_r_*, we used the equation:(1)$$S_{r}=\frac{1}{LIR}\left|\frac{d\left(LIR\right)}{dT}\right|\cdot 100\%\;$$

The *S_r_* functions obtained from *LIR* curves are represented in Figure 7.

The obtained *S_r_* values are quite competitive compared to our previous work [12]. As we mentioned above, for 0.5 and 1.0 mol.% single-doped Nd^3+^:YF_3_ the luminescence decay time of ^4^F_3/2_ (Nd^3+^) deceases with the temperature decrease in contrast to Nd^3+^ (0.1%):YF_3_ sample. It was suggested, that depopulation of ^4^F_3/2_ level occurs via cross-relaxation. At lower temperatures the distance between Nd^3+^ ions decreases due to thermal expansion and the efficiency of quenching by cross-relaxation increases. In the case of 0.1 mol.% Nd^3+^ concentration, the distance between neighboring Nd^3+^ ions seems to be larger and the interaction between Nd^3+^ ions does not occur. In terms of luminescence thermometry, the higher concentration (0.5 and 1.0 mol.%) of Nd^3+^ for Nd^3+^/Yb^3+^ ion pair seems to be more attractive. Indeed, the depopulation of ^4^F_3/2_ level happens via both cross-relaxation between Nd^3+^ ions and phonon-assisted energy transfer between Nd^3+^ and Yb^3+^. It provides faster change of Nd^3+^ emission intensity or decay time with temperature compared to Nd^3+^/Yb^3+^ doped samples, having 0.1 mol.% concentration of Nd^3+^. This faster change of Nd^3+^ luminescence parameters can increase temperature sensitivity of Nd^3+^, Yb^3+^:YF_3_ samples. To verify this suggestion, we synthesized Nd^3+^ (0.5 mol.%), Yb^3+^ (0.5 and 1.0, mol.%):YF_3_ nanoparticles via the same chemical method. However, in this system, further increase of Yb^3+^ content (higher than 1.0 mol.%) led to a significant increase of Yb^3+^ emission intensity and simultaneous decrease of Nd^3+^ one. This feature led to notable errors in *LIR* calculation. In turn, Nd^3+^ (1.0 mol.%):YF_3_ showed notably low luminescence intensity due to concentration quenching (it also can be seen from luminescence decay time curves). Thus, we calculated S_r_ for two Nd^3+^ (0.5 mol.%), Yb^3+^ (0.5 and 1.0, mol.%):YF_3_ samples (Figure 8).

It can be seen, that the obtained *S_r_* values are higher compared to the *S_r_* values calculated for Nd^3+^ (0.1 mol.%), Yb^3+^ (X mol.%):YF_3_ samples at lower temperatures. Probably, the additional temperature-dependent depopulation of ^4^F_3/2_ level via cross-relaxation plays a crucial role in its temperature sensitivity in 80–150 K range. As it was mentioned above, the shape of the *LIR* curves is affected by the difficult competitive energy exchange processes between the doping ions. Since, the sensitivity curves are obtained from the *LIR* curves, their shapes differ between each other as well. Note, that the presence of the singular points is related to the fact, that we use absolute values (modulus) of the sensitivity curves (Equation (1)). Hence in the area of the singular point without modulus, the S_r_ curve “goes” from positive part of XY plot to negative one or inversely. It can be seen, that the obtained S_r_ values are in the 0.1–0.4%·K^−1^ in the physiological temperature range. These results are comparable to one of the main competitor Nd^3+^, Yb^3+^:LiLaP_4_O_12_ (0.1–0.3%·K^−1^) [15]. In the case of Tm^3+^/Yb^3+^ down-conversion system operating in the biological window, the obtained S_r_ values also exceed the results obtained in [22]. It also should be noted, that the maximum S_r_ values are observed in the 80–150 K range. It can be concluded, that the synthesized phosphors are useful in cryogenic technique as well as in the space industry. The characterization of the present nanoparticles at lower temperatures is one of the next steps of the present study. Effective luminescence decay times τ_eff_ (^4^F_3/2_–^4^I_9/2_ transition, 866 nm emission) as functions of temperature for Nd^3+^ (0.5 mol.%), Yb^3+^ (X mol.%):YF_3_ and Nd^3+^ (0.1 mol.%), Yb^3+^ (X mol.%):YF_3_ are represented in Figure 9a,b, respectively.

It can be seen, that the temperature dependence of τ_eff_ of Nd^3+^ (0.5 mol.%), Yb^3+^ (X mol.%):YF_3_ is more pronounced compared to Nd^3+^ (0.1 mol.%), Yb^3+^ (X mol.%):YF_3_ sample. Indeed, for 0.5% Nd^3+^ sample τ_eff_ changes in the ~100 μs time frame in opposite to 0.1% Nd^3+^ (~50 μs). It, also, can be related to the fact, that for 0.5 mol.% Nd^3+^ depopulation of ^4^F_3/2_ level occurs via cross-relaxation and phonon-assisted energy transfer in contrast to 0.1 mol.% Nd^3+^. Finally, it can be concluded, that the Nd^3+^ (0.5 mol.%), Yb^3+^ (X mol.%):YF_3_ samples are more effective for both ratiometric and lifetime temperature sensing especially in the 80–150 K range.

## 4. Conclusions

This work was devoted to the study of thermometric performances of Nd^3+^ (0.1 or 0.5 mol.%), Yb^3+^ (X%):YF_3_ nanoparticles. Firstly, the nanoparticles were characterized via well-known physical methods. Particularly, according to the TEM image, the nanoparticles demonstrate 231 ± 8 nm average diameter. The obtained XRD patterns agree with both the simulation and the reference pattern from the Inorganic Crystal Diffraction Database of orthorhombic YF_3_ (P_nma_ space group (no. 074–0911)). Temperature sensitivity of spectral shape is related to the phonon-assisted nature of energy transfer (PAET) between Nd^3+^ and Yb^3+^. However, in the case of single-doped Nd^3+^ (0.1 or 0.5 mol.%):YF_3_ nanoparticles, luminescence decay time (LDT) of ^4^F_3/2_ level of Nd^3+^ in Nd^3+^ (0.5 mol.%):YF_3_ decreases with the temperature decrease. In turn, luminescence decay time in Nd^3+^ (0.1 mol.%):YF_3_ sample remains constant. It was proposed, that at 0.5 mol.% the cross-relaxation (CR) between Nd^3+^ ions takes place in contradistinction from 0.1 mol.% Nd^3+^ concentration. The decrease of LDT with temperature is explained by the decrease of distances between Nd^3+^ with temperature (thermal expansion phenomenon) that leads to the increase of cross-relaxation efficiency. It was suggested, that the presence of both CR and PAET processes in the studied (Nd^3+^ (0.5 mol.%), Yb^3+^ (X%):YF_3_) nanoparticles provides higher temperature sensitivity compared to the systems having one temperature-dependent process (Nd^3+^ (0.1, 0.5 mol.%), Yb^3+^ (X%):YF_3_). The experimental results confirmed this suggestion. The maximal relative temperature sensitivity was 0.9%·K^−1^ at 80 K.

## Figures and Tables

**Figure 1 materials-16-00039-f001:**
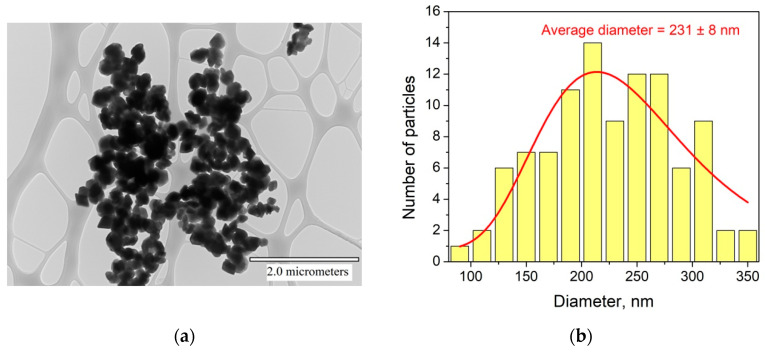
TEM image of Nd^3+^:YF_3_ particles (**a**). Size distribution histogram of Nd^3+^:YF_3_ particles (**b**) (fitting function is LogNormal from OriginPro.9.0. database).

**Figure 2 materials-16-00039-f002:**
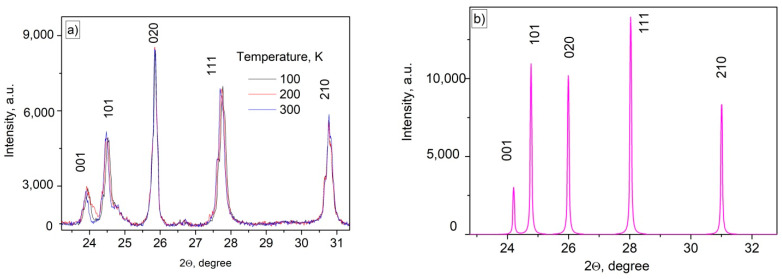
Experimental XRD patterns (**a**) of Nd^3+^ (0.5%), Yb^3+^ (1.0%):YF_3_ sample and XRD simulation of YF_3_ (**b**).

**Figure 3 materials-16-00039-f003:**
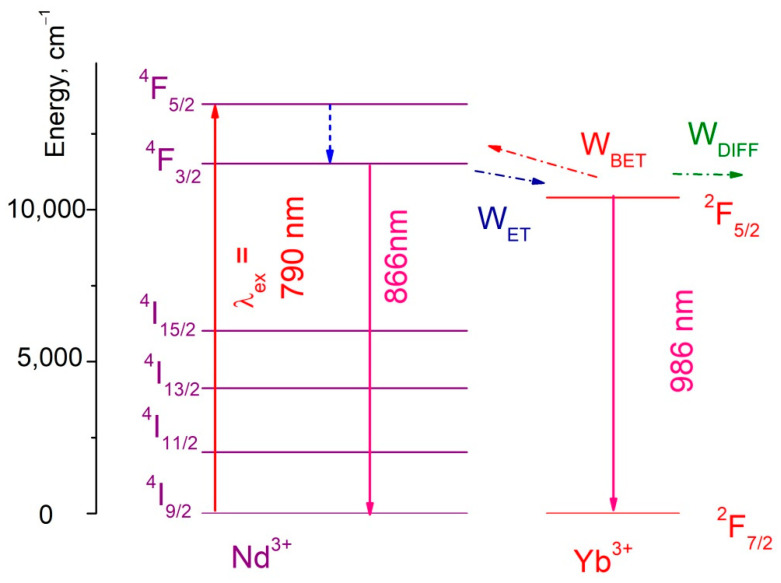
An energy level diagram of the Nd^3+^/Yb^3+^ ion pair. Here W_ET_—probability of energy transfer from Nd^3+^ to Yb^3+^, W_BET_—probability of back energy transfer from Yb^3+^ to Nd^3+^, W_DIFF_—probability of energy diffusion between Yb^3+^ ions.

**Figure 4 materials-16-00039-f004:**
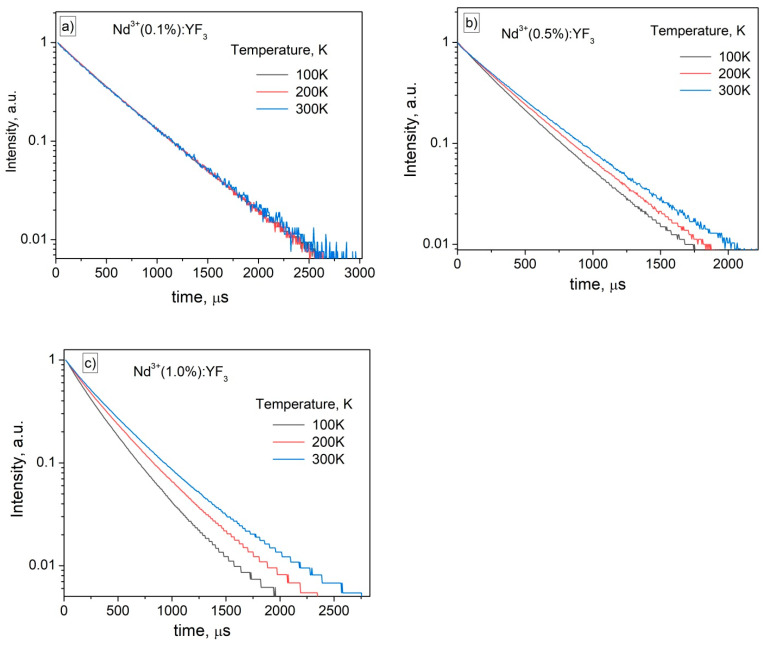
The luminescence decay time curves detected at 100, 200, and 300 K for single-doped Nd^3+^ (0.1 (**a**), 0.5 (**b**), and 1.0 (**c**) mol.%):YF_3_ nanoparticles. λ_ex_ = 790 nm (^4^I_9/2_–^4^F_5/2_ absorption band of Nd^3+^), λ_em_ = 863 nm (^4^F_3/2_–^4^I_9/2_ emission band of Nd^3+^).

**Figure 5 materials-16-00039-f005:**
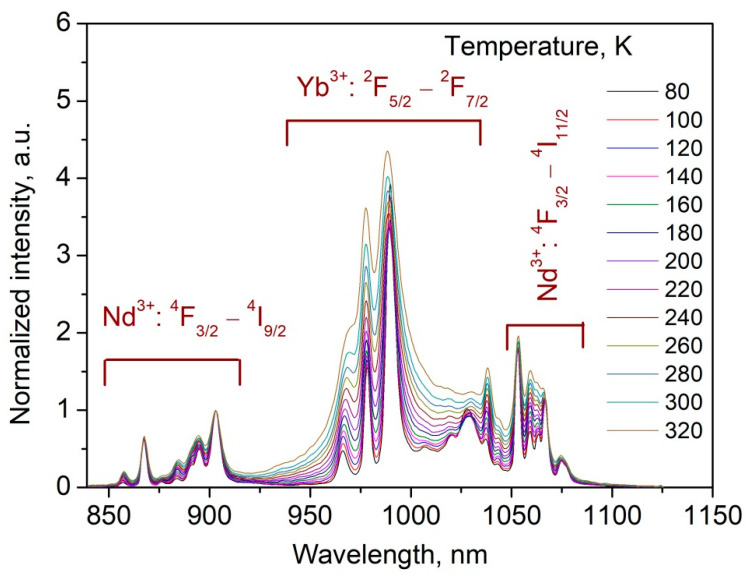
Normalized at 903 nm (Nd^3+^ emission peak) spectra of Nd^3+^ (0.1 mol.%), Yb^3+^ (1.0 mol.%):YF_3_ sample recorded in the 80–320 K temperature range. 790 nm excitation wavelength corresponds to ^4^I_9/2_–^4^F_5/2_ absorption band of Nd^3+^ ions.

**Figure 6 materials-16-00039-f006:**
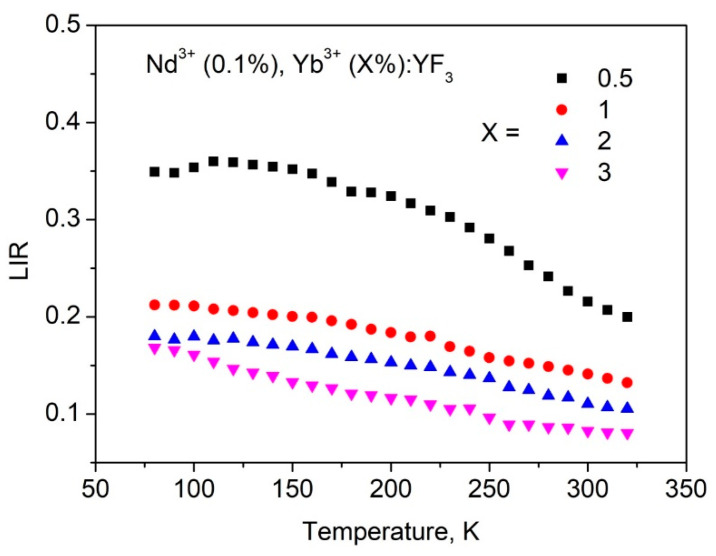
*LIR* as a function of temperature.

**Figure 7 materials-16-00039-f007:**
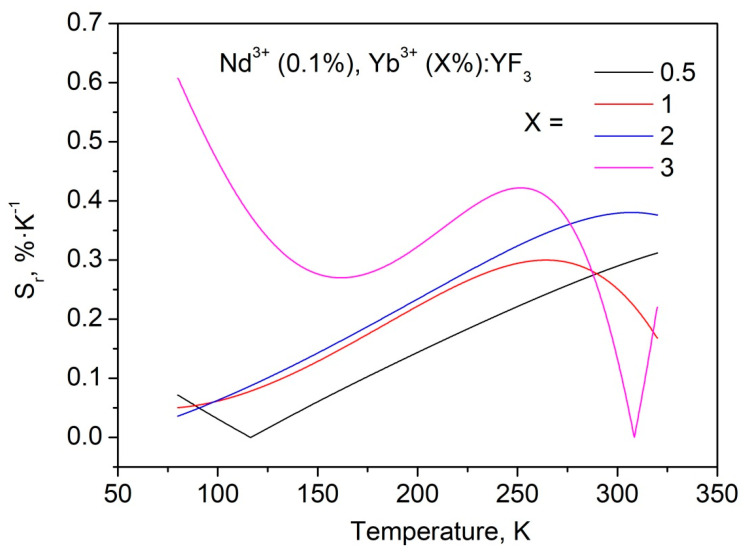
*S_r_* as functions of temperature obtained from *LIR* curves for Nd^3+^ (0.1 mol.%), Yb^3+^ (X mol.%):YF_3_ samples.

**Figure 8 materials-16-00039-f008:**
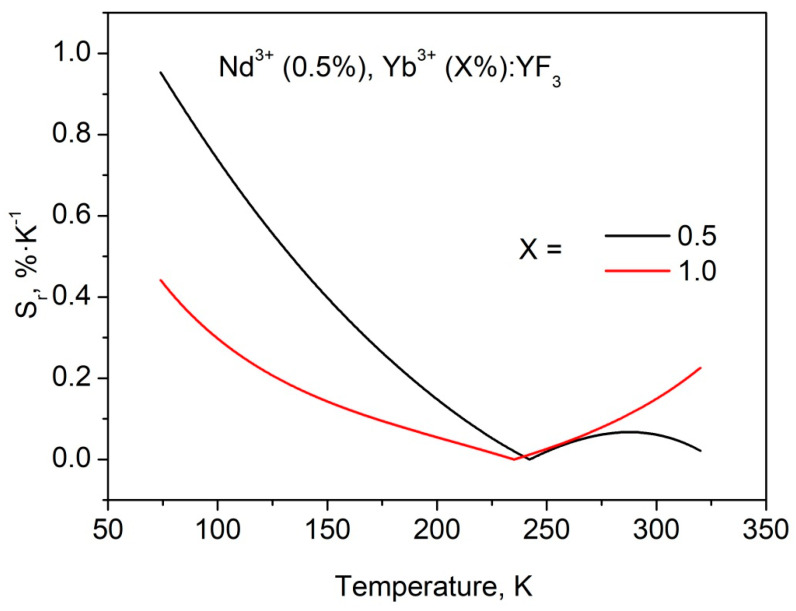
*S_r_* a functions of temperature obtained from *LIR* curves for Nd^3+^ (0.1 mol.%), Yb^3+^ (X mol.%):YF_3_ sample.

**Figure 9 materials-16-00039-f009:**
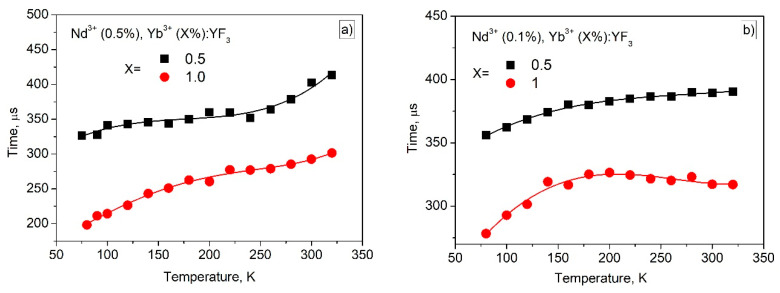
Effective luminescence decay times as functions of temperature for Nd^3+^ (0.5 mol.%), Yb^3+^ (X mol.%):YF_3_ and Nd^3+^ (0.1 mol.%) (**a**), Yb^3+^ (X mol.%):YF_3_ (**b**).

## Data Availability

Not applicable.

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
