# Peer review of "Nd3+, Yb3+:YF3 Optical Temperature Nanosensors Operating in the Biological Windows"

_materials, 2022, doi:10.3390/ma16010039_

Round 1

Reviewer 1 Report

Dear Editor, Dear Authors,

Article 'Nd3+, Yb3+: YF3 optical temperature nanosensors operating in biological windows', submitted for publication to Materials, deals with the study of the thermometric performances of Nd3+, Yb3+:YF3 nanoparticles. Particular attention is devoted to the identification of the best concentrations of Nd3+ and Yb3+ to obtain the best temperature sensitivity of the material.

The subject of this paper is interesting, but the article contains many typing errors and expressions to be corrected, some of them are listed below.

Some experimental data should be included to support the statements made in the text and some points should be clarified to be clearer to less experienced readers in this specific sector.

In particular, the abstract is poorly written and contains a number of errors that must be corrected, such as the following.

1. '... This work is devoted to studying' 

2. “…decreases with the temperature decrease...”..

3. 'The decrease of'

4. 'The maximal relative temperature...'

Errors in the Introduction and in the Main Text:

1. “... submicron...” should be corrected with “... submicrometer...”

2. {Formatting Citation}.

3. “In turn, the working in the UV, …” should be corrected with “…, working in the …”

4. “… nono-sized phosphors..” .

5. “… The tasks of the work were investigation of …” should be corrected with “The tasks of the work were to investigate … and to calculate …”.

6. “… The resulting nanoparticles were dried on-air at room temperature in a dust proof box. …” should be corrected with “… were dried in air …”

7. “ …placed in …” should be corrected with “… placed on …”

8. “in squire nanometers …”

9 “The statistics is  …” should be corrected with “… are …”

10. “ … the particle is large than 30 nm …” should be corrected with “… larger  than…”

11. “… is notably depended…” should be corrected with “… dependent…”

12. “… from lumnescence decay …”

13. “Size distribution hystogram …”

14. “… deceases with the temperature decrease …”

15. “(0.1 or 0.5 mol.%),:YF3”

16. “It can also be seen, that some XRD peaks sift toward higher angels with the temperature …”

17. “peaks sift …”

Some expressions should be modified or clarified.

Page 2 reports: “The objective of this work was to characterize the Nd3+,Yb3+:YF3 nono-sized phosphors having different concentrations of doping ions. The tasks of the work were investigation of the spectral-kinetic characteristics of Nd3+,Yb3+:YF3 nanoparticles and calculation of the main characteristics of temperature sensors.” These two statements, reported in the text, seem totally disjoint from each other, the purpose of the work must be described in greater detail and clarity.

Page 3 reports: “… before each experiment, we recorded the Nd3+ peak …We did not use the values of power density for which the changing of the peak shape was detected.” The experimental data, collected as a function of the excitation intensity, should be reported in support of the statements made, at least as S.I.

Page 4 reports: “Anyway, the size of the particle is large than 30 nm, hence, the influence of surface can be neglected [21].” This statement is unclear to a less experienced reader in the field and needs to be clarified further.

“Figure 2. Experimental XRD patterns (a) of Nd3+ (0.5%), Yb3+ (1.0%):YF3 sample and XRD simulation of YF3 (b).” In the figure, the attributions of some reported signals are missing and must be added! Furthermore, if the scales of the two figures were the same, it would be easier for the reader to compare the experimental and calculated data.

“It can also be seen, that some XRD peaks sift toward higher angels with the temperature decrease that can be related to the lattice parameters decrease. Indeed, the calculated a constant values were …” The information on which calculation method or software is used for XRD data analysis is missing.

“At higher Yb3+ concentrations, its emission intensity was much higher than Nd3+ emission that led to deterioration of the performances.” In addition, these types of data are missing and should be added as SI.

“The slight difference in LIR functions requires addition study, however, it can be suggested, that this difference related to back energy transfer from Yb3+ to Nd3+ that is different for different Yb3+ concentrations …”  This statement is unclear and poorly written, it needs to be fixed!

Figure 3. The meaning of the symbols in Figure 3 (Wbet, Wdiff, Wet) must also be added to the caption of the figure!

The behavior of the curves in figures 7 and 8 should be well described, particularly at the singular points that emerge in the graphs.

Equation (1) reports F(T). It is not clearly specified in the text to which physical quantity the quantity F(T) is correlated.

The transition:  between 4F3/2 4I9/2 (Nd3+) 221 and 2F7/2 2F7/2 (Yb3+) ” shown on page 9 is wrong and needs to be corrected!

The result obtained within this article, “The maximal relative temperature sensitivity was 0.9 %·K-1 at 80 K”, must be compared with data on other florescent systems reported in the literature, not only internal ones. Only from this comparison can we appreciate the quality and effectiveness of the synthesized nanomaterial. In particular, the maximum sensitivity is obtained at 80 K, a temperature that is not practical for applications in the biological field. Authors should comment on this important point well.

Reviewer 2 Report

In this manuscript, the authors have prepared YF3 nanoparticles (Yb3+: 0.5, 1, 2, 3 mol%) doped with different percentages of Nd3+ (0.1 or 0.5 mol%) and characterized them using transmission electron microscopy (TEM), powder X-ray diffraction techniques. Furthermore, the temperature dependency of phonon-assisted energy transfer between Nd3+ and Yb3+ using fluorescence lifetime studies and variable temperature steady-state emission techniques. The temperature sensitivity was determined to be 0.9% K-1 at 80 K. The near-infrared excitation wavelength of the energy transfer system makes it helpful to operate in biological windows.

It is interesting work and the experiments are executed in a proper way. The results are explained with sound rationale and also with the help of suitable test experiments. In my opinion, the work is suitable for publication once the authors address the below-mentioned minor aspects.

1.      Page 1: Traditional methods of measuring temperature 25 are thermocouples, thermistors, infrared cameras {Formatting Citation}. ?

2.      Page 5: XRD peaks sift toward higher angels. It should be XRD peaks shift toward higher angles.

3.      The following sentence needs to be clarified: Indeed, the calculated a constant values were 6.1300(2), 6.2109(3), and 6.3412(1) for 100, 200, and 300 K, respectively. Discuss the calculated values properly corresponding to which parameter.

4.      Figure 4 shows only the fluorescence lifetime decay profiles. It will be helpful for readers if authors also discuss the observed changes in lifetime decay in terms of decay constants.

5.      Page 6: single-to-noise ratio – it should be signal-to-noise ratio.

6.      Page 7. Provide the expansion of LIR in the main text instead of the figure caption.

Reviewer 3 Report

The study presented in this research is sound, and the results produced are interesting. But a revision is required, and after responding to the following remarks and revising the paper, the manuscript may be considered for publication.

1. The novelty of the work is missing in the introduction. Authors are suggested to include a separate paragraph discussing the novelty and importance of the present work.

2. More recent relevant literature or similar work discussion is mandatory in the introduction section, which is missing in the Introduction. Authors are suggested to add one paragraph in the introduction section by discussing the recent progress and citing similar work.

3. Authors are suggested to include a literature review on the optical, sensor, and relevant studies in a separate paragraph of the introduction section: DOIs: 10.1021/acsaelm.1c00703; 10.1016/B978-0-12-824272-8.00009-9; 10.1021/acsaelm.1c00682.

4. Reduce the similarity. Check the attached report.

5. Also, check the typos throughout the manuscript during revision submission.

Round 2

Reviewer 1 Report

Dear Editor, Dear Authors,

The various points underlined have been clearly exploited, but only one point should be still clarified:

“At higher Yb3+ concentrations, its emission intensity was much higher than Nd3+ emission that led to deterioration of the performances.” In addition, these types of data are missing and should be added as SI.

Reply. Thank you for this important comment. Undoubtedly, these data are very useful. Unfortunately, we only observed this phenomenon without stout detecting as we detected the presented in the work spectra. Probably, we can carry out these experiments by request.

If experimental data are not available, it is better to underline in the text on which basis your observation is based. How authors may affirm this point?
